# Food Insecurity Is Associated with Depression among a Vulnerable Workforce: Early Care and Education Workers

**DOI:** 10.3390/ijerph18010170

**Published:** 2020-12-29

**Authors:** Ivory H. Loh, Vanessa M. Oddo, Jennifer Otten

**Affiliations:** 1Nutritional Sciences Program, University of Washington School of Public Health, 305 Raitt Hall, P.O. Box 353410, Seattle, WA 98195, USA; jotten@uw.edu; 2Department of Health Services, University of Washington School of Public Health, 1959 NE Pacific St., P.O. Box 357660, Seattle, WA 98195, USA; voddo@uw.edu; 3Department of Kinesiology and Nutrition, University of Illinois Chicago, 1919 West Taylor St., MC 517, Chicago, IL 60612, USA; 4Department of Environmental and Occupational Health Sciences, University of Washington School of Public Health, 1959 NE Pacific St., P.O. Box 353410, Seattle, WA 98195, USA

**Keywords:** food insecurity, depression, mental health, early care and education, childcare

## Abstract

*Objective:* We aimed to explore the association between food insecurity and depression among early care and education (ECE) workers, a vulnerable population often working in precarious conditions. *Design:* We utilized cross-sectional data from a study exploring the effects of wage on ECE centers. Participants were enrolled between August 2017 and December 2018. Food insecurity was measured using the validated six-item U.S. Household Food Security Survey Module and participants were categorized as food secure (score = 0–1), low food security (score = 2–4), and very low food security (score = 5–6). Depression (defined as a score ≥ 16) was measured using the 20-item Center for Epidemiologic Studies Depression Scale-Revised. We employed a logistic regression model to examine the relationship between food insecurity and depression. All models controlled for marital status, nativity, race/ethnicity, number of children in the household, job title, weekly hours of work, education, income, and study site. *Setting:* Participants were from Seattle (40%) and South King County (26%), Washington, and Austin, Texas (34%). *Participants:* Participants included 313 ECE workers from 49 ECE centers. *Results:* A majority of participants were female, non-Hispanic White, born in the U.S., and did not have children. Compared to being food secure, very low and low food insecurities were associated with a 4.95 (95% confidence interval (CI): 2.29, 10.67) and 2.69 (95% CI: 1.29, 5.63) higher odds of depression, respectively. *Conclusions:* Policies and center-level interventions that address both food insecurity and depression may be warranted, in order to protect and improve the health of this valuable, yet vulnerable, segment of the U.S. workforce.

## 1. Introduction

In 2019, 10.5% of U.S. households (13.7 million households) experienced low or very low food security, defined as households that lack stable access to sufficient food [1]. Of U.S. households with children, 6.5% (2.4 million households) were food insecure in 2019, with very low food secure households reporting that children experienced hunger, skipped a meal, or did not eat for an entire day due to inability to purchase food [1]. Food insecurity is associated with greater depressive symptoms among both the general U.S. population [2,3,4] and a more vulnerable population: female welfare recipients [5]. Furthermore, food insecurity may contribute to children’s self-perceived psychological distress [6]. Prior studies also find high prevalence rates of both food insecurity and depression among the early care and education (ECE) workforce [7,8,9], a vulnerable population who typically earns low wages, works long hours, and lacks fringe benefits [7,8]. ECE settings include sites in which children younger than age six are cared for by a caregiver other than their parents or primary caregivers and include both center-based (e.g., childcare centers, preschools) and home-based care arrangements (e.g., nanny or babysitting in the child’s home) [8,10]. Despite the high prevalence rates of both food insecurity and depression, and the precarious work conditions faced by ECE workers in the U.S., studies have yet to explore the relationship between food insecurity and depression among this population [7].

Several prior studies have examined the relationship between food insecurity and adverse mental health outcomes [11,12,13,14,15], and most report that food insecurity is associated with a higher prevalence of depressive symptoms in the U.S. [2,3,4,5,11,16,17,18]. In particular, a 2019 meta-analysis (N = 57 cross-sectional studies) [12] reported that the odds of depression were 174% higher among those who were food insecure (compared to food secure). Similarly, a longitudinal study reported a dose–response relationship between food insecurity and depressive symptoms and poor mental health, among women at risk of or living with HIV, i.e., women experiencing more severe food insecurity had higher odds of probable depression when compared with food secure women [18]. At the same time, longitudinal analyses suggest that the relationship could be bidirectional, meaning that food insecurity is both associated with depressive symptoms and depressive symptoms are associated with food insecurity [11]. For example, a 2016 analysis using data from the Early Childhood Longitudinal Study—Birth Cohort reported that severe maternal depression was associated with a higher probability of child (79%) and household (69%) food insecurity [4].

There are several mechanisms through which food insecurity might directly be related to mental health. First, food insecurity may act as a stressor, as individuals are plagued with a constant anxiety about household food supply and worried about meeting practical needs [11,15]. Chronic stress has been found to induce physiological changes, including hyper-activation of the hypothalamic-pituitary-adrenal (HPA) axis and increased cortisol levels, that predispose individuals to depression [19]. In addition, the way in which an individual interprets their food insecurity status relative to the psychosocial environment and their social positioning may evoke shame, as well as trigger stress and adverse physiological responses [2]. Food insecurity may also be related to poor mental health by limiting one’s ability to consume a nutritious diet [2,20], provide self-care, and/or adhere to medical recommendations [11,15]. Poor self-management of chronic mental and physical health conditions may also increase an individual’s health care costs, which, in turn, can lead to additional financial strain, continued food insecurity, and limited access to health care.

Food insecurity may also indirectly harm mental health through participation in food assistance programs. The Supplemental Nutrition Assistance Program (SNAP) is the largest federal food and nutrition assistance program that supports low-income Americans [14]. Although SNAP participation is associated with a significant reduction in both prevalence and severity of food insecurity [14], participation in food assistance programs and the stigma associated with receiving food assistance have been linked to adversely affect mental health [3,5].

The ECE workforce is predominantly female and a particularly vulnerable, yet very important, segment of the U.S. workforce [7,21]. ECE jobs are often precarious or lower quality, characterized by low wages, high stress, long hours, lack of fringe benefits, and short tenure [7,21]. Compared to national averages or women of similar demographics, this workforce has been found to have unhealthier diets, lower rates of physical activity, fewer hours of sleep, and higher rates of depression and diabetes [7,8]. Moreover, this population has a high prevalence of both food insecurity and depression [7,8,9]. A study in North Carolina found that 36% of ECE workers reported clinically depressive symptoms—about five times the national rate of depression for Americans [7]. Similarly, a 2017 Arkansas workforce study reported that 33% of their sample of ECE teachers were at risk for depression, and 40% reported being food insecure [9]. A 2020 scoping review on the health status of the U.S. ECE workforce and health-promoting interventions targeting this population further validated that this workforce experiences significant mental health challenges, including stress and depression, and has a heightened chronic disease risk due to suboptimal health behaviors, regardless of ECE setting (e.g., federally funded Head Start Programs for low-income households vs. for-profit or non-profit childcare centers) and job title [22].

When examining the ECE workforce beyond the U.S., research shows that workplace stress is common among workers in other countries [22]. ECE workers in other countries, such as Singapore, also experience low pay, high turnover rate, and poor prestige [23]. The policy context for ECE varies by country. The policies, systems, regulations, practices, and culture of ECE within each country, therefore, significantly shape the health and quality of the workforce [24]. For example, the U.S. has a national recommendation on the maximum number of children per staff member based on the age of child, such as 4:1 for children up to 1 year old and 10:1 for children 2 to 5 years old, with regulations enforced by individual states [24]. The Russian Federation, on the other hand, bases its staff-to-child ratio on the available floor space of the ECE center rather than age of child. Nonetheless, there is a growing recognition among governments worldwide on the importance of investing in quality early education through the development of an effective and accessible ECE system with a well-trained and sustainable workforce [24].

The primary objective of this analysis was to explore the association between food insecurity and depression among ECE workers, as prior studies have yet to investigate this association among this high-risk population. Given the prior literature that focuses on female welfare recipients and households with children, we hypothesized that food insecurity will be associated with higher odds of depression among this vulnerable segment of the U.S. workforce. The secondary objective of this analysis was to investigate the extent to which the relationship between food insecurity and depression varied by wages and participation in food assistance programs. A better understanding of this relationship is needed to inform interventions and policies that are designed to improve the health and well-being of the ECE workforce, which may benefit the workers themselves, as well as the children whom they care for [25,26].

## 2. Methods

The cross-sectional data presented in this paper were collected as part of baseline data collection from a 2017–2020 prospective study titled “Exploring the Effects of Wage on the Culture of Health in Early Childhood Education Centers,” which explores the effects of wage on early care and education centers in Seattle and South King County, Washington, and Austin, Texas [8,27].

### 2.1. Participants, Recruitment, and Data Collection

The study population and study recruitment are described in detail elsewhere [8]. Briefly, between August 2017 and December 2018, ECE centers that served children ages 0–6 were recruited in comparable urban areas, specifically Seattle, South King County, and Austin. Seattle and South King County share similar key cost-of-living measures, including food costs and housing-cost burden [28]. Austin, TX, was also chosen as an additional comparison site to Seattle, WA, due to its similarity in cost of living, demographic characteristics, and the ECE context (Appendix A). Forty-nine (15–19%) of ECE centers that were contacted in Seattle (N = 16), South King County (N = 16), and Austin (N = 17) were enrolled. In-person worker recruitment meetings were then conducted by study staff at each enrolled ECE center. Inclusion criteria included the following: being an adult worker (≥18 years old); being employed part-time or full-time in one of the 49 ECE centers, in a position that cared for children; and being able to read and speak English. All eligible workers were invited to participate in the study. Of 504 workers who initially expressed interest in participating, 366 enrolled in the study and completed the baseline questionnaire. Of these 366 participants, 313 workers had complete case information. Participants were compensated with a $30 gift card upon completion of the baseline survey.

The baseline survey was offered online or on paper. All interested participants (n = 504) were e-mailed a link to the online survey or mailed a paper copy of baseline surveys based on their preference. Data were collected on sociodemographic characteristics (e.g., age, race/ethnicity, education, marital status, household and individual annual income, food assistance participation), employment characteristics (e.g., job title, average paid hours of work per week), and workers’ health (e.g., self-reported depressive symptoms and food security).

The Institutional Review Board at the University of Washington approved all study protocols.

#### 2.1.1. Key Exposure Variable

Our primary exposure was food insecurity, which was measured using the validated six-item U.S. Household Food Security Survey Module developed by the National Center for Health Statistics [29]. This module queries individuals about their household food situation with questions about how often in the last 12 months the participant and/or adults in his/her household ran out of food and did not have money to buy more and how often they could not afford to eat balanced meals [29]. Participants were also asked questions (yes/no) about whether they and/or adults in their household reduced the size of meals or skipped meals due to lack of money for food in the last 12 months. Affirmative answers to these questions were summed to form a household raw score (range = 0–6). Using established guidelines [29], participants were categorized into three categories: normal to high food security (score = 0 to 1), low food security (score = 2 to 4), and very low food security (score = 5 to 6). Participants categorized in the normal to high food security group are referred to as “food secure” throughout this manuscript.

#### 2.1.2. Key Outcome Variable

Our primary outcome was depression, which was measured using the validated 20-item Center for Epidemiologic Studies Depression Scale-Revised (CESD-R) [30,31]. Respondents answered questions about how often they experience nine different groups of depressive symptoms: sadness, loss of interest, appetite, sleep, thinking/concentration, guilt, tired, movement, and suicidal ideation. Each item was scored as follows: 0 (“not at all or less than one day”), 1 (“1–2 days”), 2 (“3–4 days”), and 3 (“5–7 days, or nearly every day for 2 weeks”). The overall CESD-R score is a sum of the responses to the 20 questions and ranges from 0–60 [30]. For the purposes of this analysis and using previously established cutoffs from the CESD-R, participants were coded into binary categories: clinically significant depression (score ≥ 16) and not depressed (score < 16) [30].

#### 2.1.3. Effect Measure Modifiers and Confounders

Based on prior literature [3,5,17,32], we hypothesized that the effect of food insecurity and depression may vary by wages and food assistance program participation. Food assistance participation was defined as self-reported participation in at least one food assistance program, which included Supplemental Nutrition Assistance Program (SNAP), the Special Supplemental Nutrition Program for Women, Infants, and Children (WIC), farmers market WIC program, food bank, and reduced-price or free school program. Participants receiving any food assistance were categorized together due to small sample sizes for each individual food assistance program.

Workers’ wages were collected as a continuous variable. Wage was dichotomized based on the median value for each study site (i.e., <median versus ≥median). The hourly wage median at each study site was $17.35 in Seattle, $14.08 in South King County, and $14.82 in Austin.

A directed acyclic graph (DAG) was used to identify potential confounders, which were defined as variables associated with the exposure and outcome that are not along the causal pathway [33]. Existing literature was used to support assumptions made about the role of each variable and the completeness of our DAG. From our DAG, a minimally sufficient set of confounders was identified and included as covariates in our primary model. Confounders included age (continuous), marital status (never married, now married, other), birth country (U.S. versus other), race/ethnicity (Non-Hispanic Black/African American, Non-Hispanic White, Other, Hispanic), number of children in the household (0, 1, ≥2 children), job title (center director, lead teacher or instructor, teacher or instructor, assistant teacher or instructor, other), average paid hours of work per week (continuous), highest level of education (≤high school/GED, some college/associates/ECE certificate, ≥bachelor’s degree), total household income (>$25,000, $25,000–$49,999, ≥$50,000, don’t know), and an indicator variable for study site (Seattle, South King County, Austin).

### 2.2. Statistical Analysis

We employed a logistic regression model to examine the relationship between food insecurity and depression, and controlled for the aforementioned covariates.

Two sensitivity analyses were performed for the primary model. First, sex was included as an additional covariate in the model, because 94% of our sample was female. Second, annual household income was replaced with annual individual income, as a covariate. Our primary model assumed that household income more strongly influences food insecurity and depression versus individual income.

Finally, in two separate models, an interaction term was used to assess whether food assistance participation (food assistance (yes/no) × food insecurity) and wage (median wage (below/at or above) × food insecurity) modified the association between food insecurity and depression.

All statistical analyses were conducted in STATA 13 (StataCorp., College Station, TX, USA).

## 3. Results

Our primary model included 313 ECE workers, with complete case information, from 49 ECE centers in Seattle, WA (n = 126, 40%), South King County, WA (n = 81, 26%), and Austin, TX (n = 106, 34%). Fifty-three participants, who lacked complete case information, were excluded from this analysis. Compared to participants with complete data, those excluded were more likely to be foreign-born, Hispanic, not have any children in the household, and had lower educational attainment.

Demographic characteristics of our analytic sample by food security status are presented in Table 1 (additional details presented in Appendix A). The majority of participants was female (94%), non-Hispanic White (56%), born in the U.S. (86%), and did not have children (63%). Compared with workers who reported being food secure, workers who reported low and very low food security, on average, had lower household incomes. A greater proportion of these workers also reported never having been married, having two or more child dependents, having less than a bachelor’s degree, earning wages below the site median, and participating in food assistance programs. Workers who reported that they were food secure were generally older. Of the total sample (N = 313), 72 (23%) participants were participating in food assistance programs, with 37 (12%) participants or members of their households receiving SNAP benefits.

When examining the association between depression and food insecurity, we see that mean CESD-R depression raw scores increased with higher levels of food insecurity (Figure 1). In our primary model, after controlling for covariates, very low and low food insecurities, compared to being food secure, were associated with a 4.95 (95% Confidence Interval (CI): 2.29, 10.67) and 2.69 (95% CI: 1.29, 5.63) higher odds of depression, respectively (Table 2). In the sensitivity analyses, the magnitude, direction, and statistical significance were similar when we controlled for sex (low food security Odds Ratio (OR) = 2.75; 95% CI: 1.32, 5.79, very low food security OR = 5.00; 95% CI: 2.31, 10.83) and when we controlled for household versus individual annual income (low food security OR = 3.06; 95% CI: 1.49, 6.27; very low food security OR = 5.49; 95% CI: 2.63, 11.46) (Table 3).

Finally, our study found that the association between food insecurity and depression did not vary by participation in a food assistance program (*p* = 0.71) or median site wage (*p* = 0.41). Table 4 presents estimates, stratified by food assistance program participation and wage status.

## 4. Discussion

To our knowledge, this is the first study to investigate the association between food insecurity and depression among ECE workers. Prevalence rates for food insecurity and depression in our sample were 41% and 39%, respectively, which are higher than national prevalence rates of food insecurity among American households (10.5%) [1] and of depression among U.S. females (9%) [34]. However, the prevalence rates in our sample were comparable to rates noted in recent studies with ECE samples and nursing home employees, a comparable low-wage worker population [7,9,25,32].

The association between food insecurity and depression noted in our ECE sample is consistent with that of several previous studies that look at this association within vulnerable populations and predominantly female samples [4,5,18,35]. In a recent cross-sectional analysis of the National Health and Nutrition Examination Survey (2011–2014), authors report a dose–response relationship between food insecurity and depressive symptoms among diabetic adults with an odds ratio of similar magnitude to those noted in our study [35]. A second cross-sectional analysis of low-income, diabetic participants, in King County, WA, also found that being food insecure (versus secure) was associated with almost 3-times higher odds of depression, similar to our finding for food insecure adults [36]. Results from the Women’s Employment Study similarly reported that household food insufficiency was associated with depression among low-income female welfare recipients [5]. Moreover, using longitudinal data, Tuthill et al. [18] found that women with very low food security had five times the odds of depression, compared to food secure women [18].

Regarding our secondary study objective, we did not find that food assistance participation modified the association between food insecurity and depression. These findings differ from that of some studies, which reported that the association between food insecurity and depression varied by food assistance program participation [2,3,17,37]. Some nationally representative data suggest that the association between food insecurity and emotional distress or depression was higher among SNAP participants compared to SNAP-eligible nonparticipants [2,3]. Heflin and Ziliak’s [2] longitudinal analysis also noted that the magnitude of the association between food insufficiency (i.e., food insecurity with hunger) and emotional distress was larger when participants were initially transitioning onto SNAP, possibly due to the increased stigma and participants’ inexperience in navigating the system. Similarly, Leung et al. concluded that the odds of depression, in relation to food insecurity, were higher for most SNAP participants versus SNAP-eligible nonparticipants [3]. Unlike these studies, which focused on only SNAP participation, our study examined the heterogeneity in the association by food program participation in any food assistance program, because we were not powered to solely examine SNAP participation. Only 72 sample participants received benefits from any food assistance program, and only 12% of sample participants or household members (n = 34) were enrolled in SNAP, which was much smaller than that of previous studies [2,3]. This may have been attributable to differences in SNAP eligibility requirements, as our study focused on ECE workers in only two states, rather than a nationally representative sample; our sample also mostly consisted of working, able-bodied adults without dependents (ABAWD), who must abide by specific work requirements (e.g., averaging 80 work hours/month) in order to receive SNAP benefits and are only eligible for SNAP benefits for three months in a three-year period [38,39]. Additional barriers to participation may have also played a role. In focus groups conducted with our ECE providers (N = 15), participants mentioned several barriers to participation, including stigma, individual resistance to ask for help, limited time, and having a household income that was slightly over the eligibility threshold. This is consistent with prior literature [40,41], which emphasizes the administrative hurdles associated with both applying and maintaining eligibility for SNAP, as well as the program’s inability to account for short-term income volatility among SNAP-eligible participants. In addition to differences in sampling, the absence of effect modification noted in our study may also be due to the fact that food assistance participation could be acting as a mediator, rather than an effect modifier, of the relationship between food insecurity and mental health.

Surprisingly, our study also did not find that individual-level wage modified the association between food insecurity and depression. Household financial income is consistently found to be the strongest predictor of food insecurity risk [14], and the association between financial instability and poor mental health is well documented [13,18,32]. We hypothesized that participants with the lowest wage would have a larger magnitude of effect between food insecurity and depression. One possible explanation for our findings may have to do with the narrow range of income represented in this sample. Our overall sample size was also relatively small and, therefore, we may have been insufficiently powered to detect heterogeneity in the association. Another possible explanation for our findings may be related to whether the respondent was the primary wage earner in the household and, thus, subject to higher levels of stress due to financial strain. For example, a study examining low-wage nursing home workers, which is also a highly vulnerable workforce, found that the association of depression with household financial strain and food insufficiency varied by primary wage earner status [32]. Among primary wage earner participants, the odds of depressive symptoms were found to be 3.6 times higher in relation to food insufficiency (vs. food security), whereas food insufficiency was not associated with depressive symptoms, among non-primary wage earners [32]. Our survey did not include a measure that would allow for identification of primary wage earner status.

Overall, this study suggests the need for policies and interventions that address both mental health and food insecurity in this valuable yet vulnerable workforce. Although our study focused on the ECE workforce in the U.S., the issue of low wages in this employment sector may not be unique to this country, and the availability and generosity of food assistance benefits may differ. A well-nourished and healthy ECE workforce is needed in all countries to optimize the growth and development of the next generation [24]. More children than ever are enrolled in ECE programs, and these children spend a significant amount of developmental time and may receive most of their daily nutrition in this setting [42,43]. In order to provide high-quality care and education to the children they serve, ECE workers need support in their work environments; one possible approach to help optimize their working conditions and health is through population-level strategies [25,26].

Potential policy-level strategies in the U.S. could include incorporating ECE workers as recipients of the federally reimbursed nutritious meals and snacks already served on-site to children via the Child and Adult Care Food Program or adopting legislation or provisions that improve financial security for ECE workers (e.g., tax credits, raised minimum wage [44]). Centers could also provide workers with resources and connections to nutrition education and food assistance programs, as misunderstandings and uncertainty around eligibility for food stamp benefits are fairly common, and population-specific outreach and education around food assistance participation eligibility may encourage participation [45].

Centers can also look for opportunities to create on-site programs or create synergies with existing local programs (e.g., home preparation meal kits, weekly produce markets, or food pantries). For example, Hungry Harvest partners with various public schools in Baltimore City to provide reduced-cost, recovered produce to staff, students, and families in low-income neighborhoods [46]. In Seattle and South King County, the Good Food Bags program provides a weekly subscription to subsidized, fresh produce to lower-income families through preschools, community centers, and other community partner organizations [47]. On-site programs established through a community effort [48] are not only more accessible but may also reduce the stigma associated with food assistance [45] and thereby encourage use of these resources. Finally, potential centers could include routine staff training on mental health and emotional well-being, coping skills, and stress management.

There are several limitations of this analysis. First, a cross-sectional analysis precludes our ability to infer a causal relationship between food insecurity and depression. Second, our study also included a relatively small sample of predominantly female, non-Hispanic white, and low-wage ECE workers in two states in the U.S., which limits generalizability. Third, we did not collect data on the total number of children or adults *outside of the home* who may rely upon the incomes of the ECE workers in our sample. However, an additional sensitivity controlling for the total number of people in the household produced results similar in magnitude, direction, and statistical significance to our primary specification where we only controlled for the total number of children (Appendix A). Finally, our study findings, like most survey research, are limited by selection bias. Participants who were interested and willing to complete our baseline survey may differ from non-respondents, which could bias our study results. Nonetheless, key strengths of this study include the use of validated measures and a rigorous assessment of the relationship between food insecurity and depression among a sample of ECE workers.

## 5. Conclusions

This study found that food insecurity is associated with depression among a sample of ECE workers. Considering the high prevalence rates of food insecurity and depression within this population, policies and center-level interventions that address both food insecurity and depression may be warranted in order to protect and improve the health of this valuable, yet vulnerable, workforce. A healthy ECE workforce is vital to the delivery of quality childcare.

## Figures and Tables

**Figure 1 ijerph-18-00170-f001:**
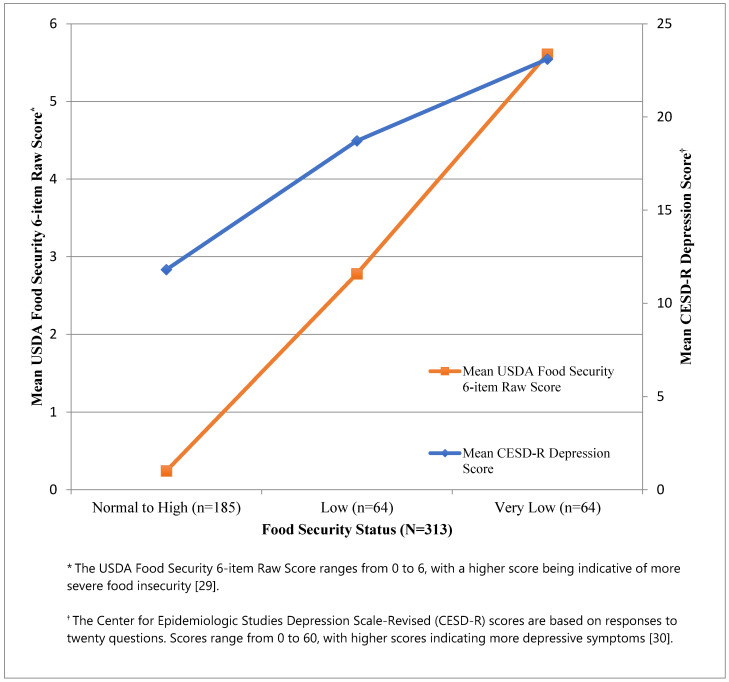
Mean USDA food security and CESD-R depression raw scores by food security status among 313 early care and education providers.

**Table 1 ijerph-18-00170-t001:** Characteristics of the study cohort of 313 early care and education (ECE) providers, by food security status, 2017.

Demographic Factor *	Food Secure (n = 185)	Low Food Security (n = 64)	Very Low Food Security (n = 64)
**Age, mean (SD)**	39.5 (13)	35.5 (13)	33.3 (12)
**Females, n (%)**	171 (93%)	59 (92%)	61 (97%)
**U.S. Born, n (%)**	155 (84%)	55 (86%)	58 (91%)
**Study Site, n (%)**			
Seattle, WA	81 (44%)	26 (41%)	19 (30%)
South King County, WA	50 (27%)	16 (25%)	15 (23%)
Austin, TX	54 (29%)	22 (34%)	30 (47%)
**Race/Ethnicity, n (%)**			
Non-Hispanic White	118 (64%)	30 (47%)	27 (42%)
Non-Hispanic Black/African-American	17 (9%)	10 (16%)	14 (22%)
Non-Hispanic, Other	27 (15%)	3 (5%)	6 (9%)
Hispanic	23 (12%)	21 (33%)	17 (27%)
**No Children (<18) in Household, n (%)**	125 (68%)	32 (50%)	40 (63%)
**Highest Level of Education, n (%)**			
≤High School or GED	17 (9%)	18 (28%)	16 (25%)
Some college, Associate’s degree, ECE certificate	65 (35%)	23 (36%)	32 (50%)
Bachelor’s degree or higher	103 (56%)	23 (36%)	16 (25%)
**ECE Job Title, n (%)**			
Center Director	20 (11%)	3 (5%)	3 (5%)
Lead Teacher or Instructor	55 (30%)	24 (38%)	28 (44%)
Teacher or Instructor	50 (27%)	14 (22%)	14 (22%)
Assistant Teacher or Instructor	41 (22%)	12 (19%)	13 (20%)
Other	19 (10%)	11 (17%)	6 (9%)
**Average Paid Hours of Work Per Week, mean (SD)**	37.7 (8)	37.2 (7)	36.5 (8)
**Individual Annual Income ($), median (25th–75th percentile)**	31,340 (25,480–37,502)	25,935 (17,160–31,221)	26,000 (15,600–30,000)
**Household income, n (%)**			
Below $25,000	22 (12%)	20 (31%)	20 (31%)
$25,000–$49,999	55 (30%)	20 (31%)	28 (43%)
$50,000 or more	96 (52%)	18 (28%)	15 (23%)
Don’t know	12 (7%)	6 (9%)	1 (2%)
**Hourly Wage in Comparison to Median Site Wage, n (%)**			
Below median	73 (41%)	34 (55%)	41 (66%)
**Depression CESD-R Score ^†^, mean (SD)**	11.8 (8)	18.7 (12)	23.1 (12)
**Non-clinical Depression ^‡^, n (%)**	138 (75%)	31 (48%)	21 (33%)
**USDA Food Security 6-item Raw Score ^§^, mean (SD)**	0.2 (0.4)	2.8 (0.8)	5.6 (0.5)
**Participates in Food Assistance Program ^◊^, n (%)**	32 (17%)	20 (31%)	20 (31%)

* Percentages provided for each demographic factor reflect proportions of participants in each food security subgroup (i.e., for each demographic factor, percentages within the same column sum to 100%). ^†^ The Center for Epidemiologic Studies Depression Scale-Revised (CESD-R) scores are based on responses to 20 questions and range from 0 to 60, with higher scores indicating more depressive symptoms [30]. ^‡^ Categorical depression was based on the CESD-R scores from 0–60 and categorized accordingly: (1) Non-clinical Depression (score = 0–16) and (2) Clinical Depression, including Major, Probable, Possible, and Sub-threshold Depression (score ≥ 16), based on previously established cutoffs [30]. ^§^ The United States Department of Agriculture (USDA) Food Security six-item Raw Scores ranged from 0 to 6, with a higher score being indicative of more severe food insecurity. Food security was categorized accordingly: (1) Normal to High Food Security (score = 0 to 1), (2) Low Food Security (score = 2–4), and (3) Very Low Security (score = 5–6) [29]. ^◊^ Out of a total of 313 samples, 37 (12%) participants or members of their households received Supplemental Nutrition Assistance Program (SNAP) benefits, 22 (7%) received Women, Infants and Children (WIC) program benefits, four (1%) received farmers market nutrition program for WIC, 13 (4%) received benefits from food bank or pantry, 24 (8%) received benefits from free or reduced school breakfast or lunch for kids, and one (0.3%) received benefits from another program.

**Table 2 ijerph-18-00170-t002:** Primary model with logistic regression estimates (odds ratios) for the association between food insecurity and depression among a cohort of 313 early care and education (ECE) providers in Washington and Texas *.

Food Security Status	Odds Ratio	95% Confidence Interval	*p*-Value
Low Food Security (N = 64)	2.69	(1.29, 5.63)	0.011
Very Low Food Security (N = 64)	4.95	(2.29, 10.67)	0.000

* Presented values were estimated using logistic regression models for the association between food insecurity and depression after controlling for age, marital status, birth country, race/ethnicity, number of children in the household, job title, average paid hours of work per week, highest level of education, total household income, and an indicator variable for study site. The reference group is participants who are food secure (N = 185).

**Table 3 ijerph-18-00170-t003:** Sensitivity analyses with logistic regression estimates (odds ratio) for the association between food insecurity and depression among a cohort of 313 early care and education (ECE) providers in Washington and Texas *.

Food Security Status	Odds Ratio	95% Confidence Interval	*p*-Value
Sensitivity Analysis 1: Primary Model + *Sex* (N = 311) ^†^			
Low Food Security (n = 64)	2.75	(1.32, 5.79)	0.009
Very Low Food Security (n = 63)	5.00	(2.31, 10.83)	0.000
Sensitivity Analysis 2: Primary Model, replacing Annual Household Income with Annual Individual Income (N = 299) ^‡^			
Low Food Security (n = 61)	3.06	(1.49, 6.27)	0.002
Very Low Food Security (n = 62)	5.49	(2.63, 11.46)	0.000

* Presented values were estimated using logistic regression models for the association between food insecurity and depression. The primary model controls for age, marital status, birth country, race/ethnicity, number of children in the household, job title, average paid hours of work per week, highest level of education, total household income, and an indicator variable for study site. **^†^** The reference group is participants who are food secure (n = 184). **^‡^** The reference group is participants who are food secure (n = 176).

**Table 4 ijerph-18-00170-t004:** Logistic regression estimates (odds ratios) for the association between food insecurity and depression among a cohort of early care and education (ECE) providers in Washington and Texas, stratified by food assistance program participation and wage *.

Food Security Status	Food Assistance Program Participation (N = 313) ^†^		Individual Wage (N = 303) ^†^	
	Yes (n = 72)	No (n = 241)	Below Site Median (n = 146) ^‡^	At or Above Site Median (n = 155)
Low Food Security	2.58 (0.41, 16.35)	2.73 (1.17, 6.38)	1.62 (0.52, 5.04)	4.11 (1.28, 13.24)
Very Low Food Security	6.77 (1.02, 44.80)	6.11 (2.48, 15.02)	6.67 (2.19, 20.38)	5.18 (1.32, 20.32)
Global *p* value	0.71		0.41	

* Presented values were estimated using logistic regression models for the association between food insecurity and depression after controlling for age, marital status, birth country, race/ethnicity, number of children in the household, job title, average paid hours of work per week, highest level of education, total household income, and an indicator variable for study site. **^†^** The reference group is participants who are food secure. **^‡^** Individuals with a job title of center director who earned below median site wage (n = 1) were not included in this model.

## Data Availability

The data presented in this study are available on request from the corresponding author. The data are not publicly available due to privacy reasons.

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
