# Peer review of "Food Insecurity Is Associated with Depression among a Vulnerable Workforce: Early Care and Education Workers"

_ijerph, 2020, doi:10.3390/ijerph18010170_

Round 1
Reviewer 1 Report
Dear authors, the requested corrections have been made.Author Response
We thank this reviewer for his/her feedback on our paper and acceptance of our revisions.
Reviewer 2 Report
Dear Authors,
I enjoyed reading the revised version of your paper.
All the best!
Author Response
We thank this reviewer for his/her feedback on our paper and acceptance of our revisions.
Reviewer 3 Report
Thank you for allowing me to review this manuscript again. The authors have adequately responded to the questions raised in the previous review. I have no additional comments to add.
Author Response
We thank this reviewer for his/her feedback on our paper and acceptance of our revisions.
This manuscript is a resubmission of an earlier submission. The following is a list of the peer review reports and author responses from that submission.
Round 1
Reviewer 1 Report
Overall, this is a well-written paper examining the association between food insecurity and depression among 313 early care and education (ECE) workers from Seattle and South King County, Washington, and Austin, Texas. The author found that food insecurity was associated with higher odds of depression among ECE workers. I have some queries which need further clarification as well as some minor comments.
- Line 31-32: Please update the prevalence of household food security in the U.S. using the most recent data released by USDA in 2020.
- Line 34-41: The target population of the current study was the early care and education (ECE) workforce, while the author mentioned other vulnerable population groups including female welfare recipients and households with children. The latter two groups seemed to be irrelevant to the study. Please justify why ECE should be studied in this study.
- Line 83-84: The primary objective was not clear regarding the study population.
- Line 94: What were the inclusion and exclusion criteria of the ECE workers. For example, do you recruit by certain age range and/or physical health status?
- Line 94-96: Were the ECE centers from the rural and/or urban areas?
- Line 97-99: The author mentioned about the data presented in this study was collected as part of baseline data from a prospective study. In total, 366 participants completed the baseline questionnaire, and 313 out of 366 participants (85.5%) completed the case information. If the data analysis in this study was only based on the baseline data collection, why did the author exclude the participants (n=53) who did not complete the case information? What was the case information about? Please justify the exclusion criteria. If the author had to exclude these participants, did they run the analysis to examine if the participants included were representative of the whole population?
- Line 170-174: When comparing the characteristics (household income, marital status, number of child in the households, education, wages, participation in food assistance programs, and age) among participants in food-secure, low-food-secure, and very-low-food-secure groups, did the author conduct any statistical analysis to compare the between-group differences? It would be good to present the p-values by FS status.
- Table 1: Was the individual annual income normally distributed across all participants included in the study? What was the range of the individual annual income? If the income was not normally distributed, the mean of individual annual income would not be the best indicator of the distribution, given the author was also presenting the hourly wage in comparison to median site wage.
- Line 199: According to table 3, the 95% CI was (2.31, 10.83).
Reviewer 2 Report
The authors have to answer a series of questions that are posed to me, in the development of this manuscript: 1.- The affiliation of the authors is incomplete. 2.- The introduction lacks a contextualization of the problem at the international level. A paragraph would be necessary to see the magnitude of the problem, and what international organizations that have reports on it, as well as its global impact. 3.- I cannot find references to the ethics committees that have analyzed this project, they are necessary in any current investigation. 4.- I would like you to explain how you have avoided location bias, and who were the researchers who passed the questionnaires in such distant locations (trainees or hired staff). 5.- The reward for filling in the questionnaires is biased, the research may be altered by complacency towards the researchers to obtain an economic benefit. 6.- How was the online questionnaire validated, and if so, were differences observed between the collection models? 7.- They should explain why the sample is feminized, since it is an element that they express in their analysis and conclusions. 8.- The discussion paragraphs should be shorter and above all, follow the order of presentation of the results presented. 9.- The bibliography numbering is wrong. You have 13/42 citations older than 5 years that you should update. Check for important formatting errors in referenced citations.Reviewer 3 Report
Dear authors,
I thoroughly read your paper. Please, find my comments below:
Introduction:
A self-reference to your conference paper is missing:
https://www.researchgate.net/publication/341740257_Food_Insecurity_Is_Associated_with_Depression_Among_a_Vulnerable_Workforce_Early_Care_and_Education_Workers#fullTextFileContent
Why do you refer to the "early life stress hypothesis"? This is not relevant for the target group (they are in their 30ies)
Please, add a separate section on the vulnerability of ECE workers
Please, insert a section on SNAPS
Results:
This chapter is a series of tables and lacks text bridges.
Section 2.1 (Profile of participants): Please provide information on their participation in SNAPS.
Table 1 is very extensive. Please, provide an excerpt of the most relevant information in the text and shift the table in whole length to the annex.
Discussion:
This section lacks structure and therefore is difficult to read.
Please, point out the identified similarities and differences of the three settings, for example by using tables.
Formatting:
Please, edit the citations in the text (see “Information for Authors”).
All the best.
Reviewer 4 Report
Thank you very much for giving me the opportunity to read and review this manuscript. I think it is an important piece of work that highlights the risks of ECE workers' working conditions and the need to implement interventions.
I think the manuscript has an excellent presentation, the methods are described in detail and clearly, and the results and the discussion allow the reader to understand the magnitude of the problem. It is appreciated that the authors include comments that relate some unexpected results to the limitations of the study.
There is only one issue that I think could be improved. The authors say in the conclusion that "This study finds that food insecurity is associated with depression." However, this association is already established in previous studies, especially in samples of vulnerable women, as the authors themselves point out in the introduction. Therefore, I would recommend that the authors emphasize in the conclusions section what is the concrete contribution of this work to what is already known on the subject. Perhaps the key is in the first sentence of the discussion.
Regarding questions of form, I would just like to indicate that the list of bibliographic references is double numbered.
Congratulations to the authors for such excellent work.